# Transcriptome Analysis of Intermittent Light Induced Early Bolting in Flowering Chinese Cabbage

**DOI:** 10.3390/plants13060866

**Published:** 2024-03-17

**Authors:** Caizhu Hu, Dongyu Sun, Jinhui Yu, Mengqing Chen, Yanxu Xue, Jinmiao Wang, Wei Su, Riyuan Chen, Ali Anwar, Shiwei Song

**Affiliations:** College of Horticulture, South China Agricultural University, Guangzhou 510642, China; cz-hu@outlook.com (C.H.); sundongyu@stu.scau.edu.cn (D.S.); yujh97@163.com (J.Y.); 20223137140@stu.scau.edu.cn (M.C.); xueyanxu@stu.scau.edu.cn (Y.X.); wangjinmiao@stu.scau.edu.cn (J.W.); susan_l@scau.edu.cn (W.S.);

**Keywords:** flowering Chinese cabbage, intermittent light, RNA sequencing, bolting, flowering

## Abstract

In flowering Chinese cabbage, early booting is one of the most important characteristics that is linked with quality and production. Through fixed light intensity (280 μmol·m^−2^·s^−1^) and fixed intermittent lighting in flowering Chinese cabbage, there was early bolting, bud emergence, and flowering. Moreover, the aboveground fresh weight, blade area, dry weight of blade, and quantification of the leaves in flowering Chinese cabbage were significantly reduced, while the thickness of tillers, tillers height, dry weight of tillers, and tillers weight were significantly increased. The chlorophyll contents and soil–plant analysis and development (SPAD) value decreased in the early stage and increased in the later stage. The nitrate content decreased, while the photosynthetic rate, vitamin C content, soluble sugar content, soluble protein content, phenolic content, and flavonoid content increased, and mineral elements also accumulated. In order to explore the mechanism of intermittent light promoting the early bolting and flowering of ‘49d’ flowering Chinese cabbage, this study analyzed the transcriptional regulation from a global perspective using RNA sequencing. A total of 17,086 differentially expressed genes (DEGs) were obtained and 396 DEGs were selected that were closely related to early bolting. These DEGs were mainly involved in pollen wall assembly and plant circadian rhythm pathways, light action (34 DEGs), hormone biosynthesis and regulation (26 DEGs), development (21 DEGs), and carbohydrate synthesis and transport (6 DEGs). Three hub genes with the highest connectivity were identified through weighted gene co-expression network analysis (WGCNA): *BrRVE*, *BrLHY*, and *BrRVE1*. It is speculated that they may be involved in the intermittent light regulation of early bolting in flowering Chinese cabbage. In conclusion, intermittent light can be used as a useful tool to regulate plant growth structure, increase planting density, enhance photosynthesis, increase mineral accumulation, accelerate growth, and shorten the breeding cycle.

## 1. Introduction

Flowering Chinese cabbage (*Brassica campestris* L. ssp. *chinensis* var. *utilis* Tsen et Lee) is a highly important vegetable in the southern region of China [1]. It has a crisp and tender texture, unique flavor, and rich nutritional content, making it highly favored by consumers [2]. The cultivation of flowering Chinese cabbage has been popularized nationwide, making significant contributions to meeting the year-round demand for vegetables [3]. Flowering Chinese cabbage is a long-day plant [4], and its growth process mainly includes germination, bolting, bud emergence, flowering, etc., with bolting being a critical turning point in the morphological differentiation of vegetables [5]. The transition from vegetative to reproductive growth in flowering Chinese cabbage is regulated by various factors, including the photoperiod, temperature, water, and nutrients [6]. Hence, the photoperiod plays a crucial role in its growth and development. However, current research mainly focuses on exploring the effects of different light quality, intensity, and artificial lighting on flowering Chinese cabbage [7,8,9]. In addition, research on regulating high-yield and high-quality flowering Chinese cabbage, accelerating growth, and shortening breeding cycles is also relatively insufficient.

Light is one of the important environmental factors for the growth and development of vegetables in facilities [10], while the characteristics and regulation of the light environment have become a hot topic in greenhouse horticulture research [11]. Previous studies showed that intermittent light is a periodically flickering light that controls the opening and closing of the light source through specific devices or systems, thereby regulating the photoperiod of plants and interfering with their normal growth and development [12]. In fact, under natural conditions, the light perceived by the lower leaves of a plant community is not continuous but is produced by the dappled light spots of the upper leaves or the intermittent light source, generated by the swinging of the upper leaves [13]. These discontinuous light sources have a positive effect on the growth status of plants. As early as 1905, researchers used a rotating disk to reduce the light intensity by 25% and found that the photosynthetic rate did not decrease, thus exploring the study of intermittent light for the first time [14]. After further research, we have found that algae respond very positively to intermittent light [15,16]. The study results show that intermittent light can significantly promote the growth rate and light energy utilization efficiency of algae, thereby increasing yield [17]. In addition, through experiments on vegetables [18], fruit trees [19], and other plants, we have also confirmed that intermittent light at a certain frequency can promote plant growth and development, accelerate substance synthesis, and increase yield. Furthermore, properly increasing the duty cycle can also improve the quality of plants [20]. Therefore, if we can properly set the frequency or period of pulse light to match the photosynthetic reaction time of plants, it can not only reduce light energy waste and improve light energy utilization efficiency, but also avoid light inhibition and promote plant growth.

The biological clock is a heritable regulatory mechanism that is widely present in various organisms, helping them adapt to the diurnal and seasonal changes in the environment [21]. It plays a regulatory role in plant photosynthesis, growth and development, and flowering time [22]. However, intermittent light may interfere with the transmission of biological clock signals, leading to early flowering in plants [23]. Several transcription factors and proteins related to circadian rhythm regulation have been discovered in *Arabidopsis* [24]. The research found that the core of the circadian clock is composed of three genes: *TOC1*, *LHY*, and *CCA1*. They form multiple transcriptional feedback loops to ensure the normal operation of the circadian clock [25]. During the morning, the expression levels of *CCA1* and *LHY* are highest. They bind to the promoter of *TOC1*, inhibiting its expression, and their own expression levels also continue to decrease. During the night, *TOC1* accumulates continuously and reaches its peak, inhibiting the transcription of *LHY* and *CCA1*, forming a morning negative feedback loop [26]. In the afternoon, *RVE8*, *RVE4*, and *RVE6* recruit transcriptional coactivators and form complexes with LNKs, promoting the transcription of *PRR9*, *PRR5*, *TOC1*, *GI*, *LUX*, and *ELF4* [27]. During the night, *TOC1* accumulates to its peak, inhibiting the expression of *CCA1*, *LHY*, *LUX*, *ELF4*, *ELF3*, and *GI* [28]. *TOC1* interacts with CHE to inhibit the transcription of *CCA1* [29]. *TOC1* accumulated during the night inhibits the expression of *GI*, while GI in turn activates *TOC1* and forms a night negative feedback loop [30].

This study explored the effects of intermittent light on the growth and flowering of flowering Chinese cabbage through transcriptome analysis and revealed the potential molecular mechanisms behind its early bolting. These research findings contribute to optimizing the growth cycle of flowering Chinese cabbage, achieving early harvest, and increasing market competitiveness. At the same time, the results also provide scientific basis for the cultivation and management of flowering Chinese cabbage. Through in-depth research on intermittent light, we can better understand the growth law of flowering Chinese cabbage and use this knowledge to optimize cultivation techniques and improve yield and quality.

## 2. Results

### 2.1. The Effect of Intermittent Light on the Growth and Development of Flowering Chinese Cabbage

As a leafy vegetable, flowering Chinese cabbage is very sensitive to environmental influences which can decline plant growth and production. Among these environmental hurdles, light is one of the most important factors that affect plant growth. In this study, we investigate the involvement of intermittent light on flowering Chinese cabbage. We found that intermittent light treatment (the lighting mode that flashes once per second for 24 h) had a role in promoting the bolting, bud emergence, and flowering of ‘49 d’ flowering Chinese cabbage (Figure 1). By calculating the bolting rate, bud emergence rate, and flowering rate, which were more than 50%, we found that intermittent light treatment had a significant effect at 6 d, 9 d, and 12 d earlier than the CK (Figure 1a–c). Based on these results, we set four stages, namely S1 (two-leaf stage in the treatment and control groups, 11 d), S2 (treatments with bolting rate over 50% and control group without bolting, 17 d), S3 (treatments with budding rate over 50% and control group with bolting, 26 d), and S4 (treatments with flowering rate over 50% and control group with budding, 32 d). At the same time, we selected representative plants from each stage (Figure 1d). These results confirm the promoting effect of intermittent light on the entire growth stage of ‘49 d’ flowering Chinese cabbage, with significant changes in bolting occurring in the S2 stage.

### 2.2. The Effects of Intermittent Light on the Morphological Indexes on Flowering Chinese Cabbage

To investigate the promoting effect of intermittent light on the growth and development of flowering Chinese cabbage, we further measured four horticultural traits at four stages (Figure 2). The results showed that the treated group had a significantly reduced aboveground fresh weight, blade area, dry weight of blade, and quantification of leaves compared to the CK in all four stages (Figure 2). The differences in aboveground fresh weight were significant in the S2 and S4 stages. The differences in blade area were significant in all stages. The differences in dry weight of blade were significant in the S3 and S4 stages. Only in the S4 stage was there a significant difference in the quantification of leaves.

As for the tillers, in the three stages (no obvious tillers were observed in S1), the tiller’s height, dry weight of tiller’s, and tiller’s weight in the treatments were higher than those in the CK (Figure 2). The tiller’s height showed significant differences in all stages; thus, the dry weight of tillers was significantly different in the S2 and S3 stages. And the data of the treatments in the S3 stage are very similar to the data of the CK in the S4 stage. Likewise, the tiller weight showed significant differences in the S3 and S4 stages, while the thickness of tillers showed different changes in the S4 stage compared to tiller height. The thickness of tillers in the treatments were higher than that in the CK in the S2 and S3 stages, but lower in the S4 stage, and there were significant differences in the S2, S3 and S4 stages. These findings suggest that intermittent light promotes the development of flowering Chinese cabbage tiller tissue without increasing the blade area or quantification of leaves and do not affect the final yield.

### 2.3. Intermittent Light Effects on Chlorophyll Content

As mentioned above, the intermittent light can promote the early bolting and flowering of flowering Chinese cabbage. To investigate more, we measured the chlorophyll content during the four stages (Figure 3). The results showed that chlorophyll a, chlorophyll b, carotenoid, and chlorophyll (a+b) content in the CK increased significantly and then decreased, reaching the maximum values in the S2 stage. In contrast, the treatments showed a continuous increase in these contents. There were significant differences between the CK and the treatments in chlorophyll a content and chlorophyll b content in the S2, S3 and S4 stages. There were significant differences between the CK and the treatments in carotenoid content and chlorophyll (a+b) content in the S2, S3 and S4 stages. However, the chlorophyll a/b ratio showed a decreasing trend in both the CK and the treatments, with the ratio in the treatments higher than the CK in the S1 and S2 stages, and lower than the CK in the S3 and S4 stages. There were significant differences in the S3 and S4 stages. These findings suggest that intermittent light affects the accumulation of chlorophyll in flowering Chinese cabbage at different stages.

### 2.4. Intermittent Light Effects on Photosynthetic Characteristics and Chlorophyll Fluorescence Characteristics

The SPAD values of both the CK and the treatments show an upward trend, but the values of the CK are always higher than those of the treatments (Figure 4). There are significant differences between the two groups in all stages. In terms of net photosynthetic rate, both the CK and the treatments show a trend of first increasing and then decreasing, reaching their maximum values in stage S2. The net photosynthetic rate of the treatments was higher than that of the CK in the S1, S2, and S3 stages, and there were significant differences in the S1 and S2 stages. As for S4 stage, the net photosynthetic rate of the treatments is slightly lower than that of the CK.

The Fv/Fm values of the CK and light treatments showed a gradual increase trend (Figure 4). In the S1, S2, and S3 stages, the Fv/Fm values of the treatments were slightly higher than those of the CK, and there was a significant difference between the two groups in the S3 stage. However, in the S4 stage, the Fv/Fm values of the CK were significantly higher than other treatments. As for the qP values, the CK showed an upward trend while the treatments exhibited a downward trend. Additionally, the treatments were higher than the CK in the S1, S2, and S3 stages, whereas the CK was higher than the treatments in stage S4. There were significant differences in the S1, S2, and S4 stages. The Y(NPQ) values of the CK remained basically unchanged, while the treatments showed a downward trend. In the S1, S2, and S3 stages, the Y(NPQ) values of the treatments were higher than that of the CK, and there were significant differences in the S1, S2, and S3 stages. The Y(NO) values of the CK and the treatments showed a downward trend, with the values of the treatments being higher than that of the CK, and there were significant differences between the two groups in the S1 and S2 stages. The ETR values of the CK and the treatments also showed an upward trend, but the difference was not obvious, with the CK always higher than the treatments, and there were significant differences in the S1, S2, and S3 stages.

These findings indicate that intermittent light can enhance the photosynthetic capacity of flowering Chinese cabbage and play a positive role in promoting growth and development. At the same time, intermittent light can also accelerate the recovery process of damage caused during the seedling stage.

### 2.5. Intermittent Light Effects on Nutritional Accumulation, and Antioxidant Enzyme Activities

The nutrient content and antioxidant capacity of flowering Chinese cabbage leaves and tillers under intermittent light treatment are presented in Figure 5. The results in the figure show that the tillers samples in the S1 stage were obtained after removing the leaves and cotyledons. The nitrate contents in leaves and tillers under intermittent light treatments were lower than the CK, with significant differences in the leaves during the S2, S3, and S4 stages, and significant differences in the tillers during the S1, S3, and S4 stages. The vitamin C content in the leaves and tillers of the treatments were always higher than that of the CK, with significant differences in the leaves during the S2, S3, and S4 stages, and significant differences in the tillers all stages. The soluble sugar and soluble protein content of the leaves and tillers show different trends. In leaves, the soluble sugar content of the CK and treatments first increases and then decreases, reaching the highest point in the S2 stage. There were significant differences in all stages. The tillers show an increase, and treatments are higher than the CK. There were significant differences in all stages.

The soluble protein content in the leaves shows a decreasing trend in both the CK and treatments. The CK was higher than the treatments in the S1 stage by a significant difference and then lower than the treatments. There were significant differences in the S2 and S4 stages. The trend in the tillers first increases and then decreases. The CK and treatments reach the highest point in the S3 and S2 stages, respectively. The CK are higher than the treatments in the S1 and S4 stages, and lower than the treatments in the S2 and S3 stages, with a significant difference in the S2 stage.

Throughout the entire stage, the phenolic content of the CK and treatments showed a decreasing trend in the leaves, with the phenolic content of the treatments consistently higher than that of the CK. There was a significant difference between the two groups in the S2 stage. In tillers, the phenolic content showed an increasing trend, with treatments consistently higher than the CK. There were significant differences between the two groups in all stages. The flavonoid content in the leaves and tillers showed a consistent increasing trend, with the treatments consistently higher than the CK. In the leaves, there were significant differences in the S2, S3, and S4 stages. In the tillers, there were significant differences in all stages.

The DPPH values of the leaves did not show significant changes. In the S1 stage, treatments were higher than the CK, but then it was lower than the CK. There was a significant difference in the S4 stage. On the other hand, the DPPH values of tillers showed an increasing trend. The values of the treatments were always higher than the CK, with significant differences in all stages. There were significant differences in the FRAP values between the leaves and tillers. Treatments in the leaves consistently had higher values than the CK, with significant differences in the S3 and S4 stages. However, in the tillers, treatments were higher than the CK in the S1 stage, with significant differences. Then, it was lower than the CK in subsequent stages, with significant differences in the S3 and S4 stages.

The results indicate that while intermittent light promotes the growth of flowering Chinese cabbage, it also improves the taste, quality, and antioxidant capacity.

### 2.6. The Effect of Intermittent Light on the Mineral Elements

Mineral uptake and accumulation play a key role in plant growth and development. Therefore, we investigated the role of intermittent light treatment in flowering Chinese cabbage nutrients at the S2 stage (Figure 6). The results showed that the content of N, K, Ca, S, and Zn elements in the treatments were significantly higher than that of the CK, with particularly significant differences. However, the contents of Fe element in the treatments was lower than that of the CK, with significant differences. The content of P and Mg elements was slightly lower than that of the CK, but the differences were not significant. Thus, it was speculated that the intermittent light and the mineral elements in flowering Chinese cabbage affected the accumulation of chlorophyll and plant growth.

### 2.7. Identification of DEGs in Response to Intermittent Light

To investigate the effects of intermittent light on the differential gene expression of ‘49 d’ flowering Chinese cabbage in early bolting, we collected flowering Chinese cabbage inflorescence tissue samples. A total of 27 RNA-Seq libraries were constructed. We obtained a total of 3.75–5.40 × 10^7^ clean reads. The average GC content of the libraries was 47.31%, with Q30 values of 92.98%, respectively. We aligned the clean reads of each sample to the reference genome, and the alignment efficiency ranged from 87.67% to 89.22% (Appendix A). These results suggest that the illumina sequencing data are reliable and can be further analyzed.

The sequencing results showed that the number of genes detected in each library ranged from 3.08–3.35 × 10^4^, accounting for 63.68% to 69.35% (Figure 7). Based on gene expression data, we performed principal component analysis (PCA), which showed consistency among samples and experimental variation (Figure 7 and Appendix A). The sample clustering analysis revealed a high degree of correlation between the S1-1s and S2-CK, indicating that the S1-1s and S2-CK have similar states and suggesting that intermittent light can advance the bolting time by one stage.

We used DESeq2 for pairwise comparison to identify differentially expressed genes. In the nine groups, we found a total of 17,086 differentially expressed genes (DEGs) (Figure 7e). We performed co-clustering analysis on these DEGs in the nine sample groups using STEM software, which resulted in 40 expression patterns (with 20 patterns for S0 vs. the 1s group, and 20 patterns for S0 vs. the CK). Each group contained five significant expression patterns, including patterns 18, 19, 0, 16, and 2, as well as patterns 0, 18, 19, 16, and 11 (Figure 7f,g). Specifically, pattern 18 included 3778 and 3229 DEGs in the two groups, showing an initial upregulation, followed by a stable phase, and finally a downregulation, indicating an overall increasing trend with the increase in days of intermittent light exposure. Pattern 19 included 3557 and 2953 DEGs, showing an upregulation trend with the increase in days of intermittent light exposure. Pattern 0 included 2822 and 3433 DEGs, showing a downregulation trend with the increase in days of intermittent light exposure.

### 2.8. GO Enrichment Analysis and KEGG Analysis of Intermittent Light

The DEGs of the early bolting of ‘49 d’ flowering Chinese cabbage induced by intermittent light were used for Go and KEGG pathway analysis. To explore the enrichment analysis, we designated three key comparison groups: S1-CK vs. S1-1s, S1-1s vs. S2-CK, and S2-CK vs. S2-1s, and identified 2050, 2050, and 1735 non-redundant DEGs, respectively (Figure 8). Among the three groups, 1658 upregulated genes, 1615 downregulated genes and 1167 downregulated genes, respectively, occupy a significant proportion. In addition, we conducted a detailed analysis of the functions of all DEGs through GO (Appendix A) and KEGG (Appendix A) classifications.

The three comparative groups related to early bolting, significant numbers of DEGs enriched in molecular function, cellular component, and biological process, including GO terms such as “cellular components assembly involved in morphogenesis”, “pollen wall assembly”, “rhythm process”, and “S-glycoside metabolic process”. Moreover, in the KEGG pathway enrichment analysis, differentially expressed genes are mainly enriched in pathways such as “metabolic pathways”, “glucosinolate biosynthesis”, “circadia rhythm-plant”, and “starch and sucrose metabolism”. Based on these results, it can be concluded that DEGs related to morphogenesis, secondary metabolite biosynthesis, plant circadian rhythm, and starch and sucrose metabolism may be the worthiest of further research.

### 2.9. Intermittent Light Conditions, the DEGs Exhibit a Pattern of Enriched Pathways

In order to study the molecular adaptability of ‘49 d’ flowering Chinese cabbage to intermittent light, we analyzed the expression patterns of DEGs in key pathways (Figure 9). The results showed that a total of 34 genes were associated with light response. Among them, 6 genes (*BraA05g024380.3C-ELIP1, BraA03g040550.3C-ELIP1*, *BraA07g019510.3C-RBCS*, *BraA04g014860.3C-RBCS*, *BrBCA1*, *BrPSBO2)* were related to photosynthesis, 15 genes (*BrNPF6.3*, *BrCAT2*, *BrRVE2*, *BrLHY*, *BrRVE8*, *BrCOL1*, *BrRVE1*, *BrGATA3*, *BrLNK3*, *BrGRDP2*, *BrSOC1*, *BrPAR2*, *BrJMJ30*, *BrCCR1*, *BrLIR1*) were related to circadian rhythm, and 13 genes (*BrRCA*, *BrHYH*, *BrBBX25*, *BrATH1*, *BrRPT2*, *BrRAX2*, *BrBCAT4*, *BrLSH2*, *BraA03g030180.3C-UNE10*, *BraA03g025630.3C-UNE10*, *BrPNSB1*, *BrHPR*, *BrCYP707A2*) were related to light stimulation.

In the related pathways of plant hormone biosynthesis and regulation, the gene expression patterns show a complex situation. This study identified 27 DEGs, including 5 auxin response factors (*BrARF22*, *BrARF14*, *BrGER1*, *BrCYP705A22*, *BrXTH22*), 2 gibberellin synthesis genes (*BraA02g024710.3C-GA2OX1*, *BraA07g041570.3C-GA2OX1*), 3 cytokinin response factors (*BrIPT1*, *BrARR20*, *BrRAP2-3*), 4 jasmonic acid genes (*BrJMT*, *BrPAP2*, *BrLOX2*, *BrUBP12*), and 13 abscisic acid genes (*BrZFP8*, *BrDTX43*, *BrMYB96*, *BrCYP707A3*, *BrPLP3*, *BrABCG31*, *BrGINT1*, *BrFBA2*, *BrWSD1*, *BrAFP4*, *BrEXO70B1*, *BrBGLU18*, *BrNCED9*).

We have also identified 21 genes related to development, among which 15 genes (BrNAC025, BrBHLH91, BrSWEET8, BraA03g053680.3C-ABCG9, BrACA9, BraA01g018140.3C-ABCG9, BrEXL6, BraA09g011690.3C-SHT, BraA06g029650.3C-SHT, BrJGB, BraA08g028990.3C-GRP17, BraA10g029750.3C-GRP17, BraA02g020900.3C-KTI2, BrNAS4, BraA02g020890.3C-KTI2) play important roles in regulating pollen and development, 3 genes (BrSDC, BrFTIP3, BrPSK6) are associated with apical meristem tissue, and 3 genes (BrSCL29, BrNAC029 BrGRDP2) are related to flower development.

Finally, we identified 6 genes (*BrMSSP3*, *BrBGLU20*, *BrNSP2*, *BrPPT1*, *BrPLT5*, *BrBGLU47*) that are associated with carbohydrate synthesis and transport.

### 2.10. Validation of DEGs Expression using QRT-PCR, and 14 Key Genes Related to Circadian Rhythm and Flowering Was Selected for Quantitative Analysis

To verify the accuracy of the transcriptome analysis results, we randomly selected eight DEGs for confirmation using quantitative real-time reverse transcription PCR (qRT-PCR) (Figure 10). The expression profiles of the selected genes as revealed using qRT-PCR data showed similar trends to those obtained through sequencing. Subsequently, we conducted a linear regression analysis of the fold changes in gene expression between RNA-Seq and qRT-PCR. The linear regression results showed a positive correlation (R^2^ = 0.8108), indicating the reliability of the transcriptome analysis using RNA-Seq.

To investigate the effects of intermittent light on the early bolting of flowering Chinese cabbage, we further analyzed the genes with the highest number of DEGs in the light-related group. We focused on 15 key genes related to circadian rhythm and flowering. Among these 15 key genes, we identified three flowering integrator factors: *BrSOC1*, *BrCOL1*, and *BrLHY*. Among them, the expression levels of *BrSOC1* in CK and 1s showed an upward trend, but the trend line of CK was consistently lower than that of 1s. For *BrCOL1* and *BrLHY*, the trend line of CK remained relatively stable, while the trend line of 1s was above CK and showed a downward trend. By observing the relative expression levels of *BrLNK3*, *BrJMJ30*, *BrCAT2*, and *BrCCR1* genes, it can be observed that they show similar changes as *BrSOC1*. Similarly, the relative expression level changes in *BrRVE8* and *BrRVE1* genes show a similar trend as *BrCOL1* and *BrLHY*. The expression level of *BrGRDP2* showed an increasing trend in both the CK and treatments, but the change in the CK was higher than that of the treatments, and the trend line of the CK was above that of the treatments. The change trends of *BrNPF6.3* and BrLIR1 were opposite. In *BrNPF6.3*, the CK showed an increasing trend, while treatments showed a decreasing trend, with the CK initially lower than treatments and then higher. In *BrLIR1*, the CK showed a decreasing trend, while treatments showed an increasing trend, with the CK was initially higher than the treatments and then lower. The change trends of *BrRVE2* and *BrGATA3* showed that both the CK and treatments initially showed a decreasing trend. In *BrRVE2*, the CK was lower than the treatments, while in *BrGATA3*, the CK was higher than the treatments.

In conclusion, the expression levels of eight genes increased with the increase of intermittent light treatment, while the expression levels of another seven genes decreased with the increase of intermittent light treatment. Furthermore, during the intermittent light treatment process, the expression levels of four genes remain unchanged, while the expression levels of two genes are in the opposite direction to the treatment.

### 2.11. Co-Expression Modules Promoting Bolting in Flowering Chinese Cabbage under Intermittent Light

To investigate the dynamic changes in gene expression during the early bolting process of ‘49 d’ flowering Chinese cabbage, we analyzed 4278 DEGs in four stages (S1-CK, S1-1s, S2-CK, and S2-1s) using weighted gene co-expression network analysis (WGCNA). With a power value of 8, we identified 18 co-expression modules, with the number of genes in each module ranging from 22 (module 17, MM.saddlebrown) to 1149 (module 12, MM.turquoise) (Figure 11).

Through module phase correlation analysis, we found that there is a strong correlation between 11 pairs of modules, with a correlation coefficient (PCC) of ≥0.85. They are M1 (MM.green) vs. M2 (MM.purple), M2 (MM.purple) vs. M4 (MM.skyblue), M3 (MM.darkred) vs. M18 (MM.salmon), M5 (MM.lightgreen) vs. M6 (MM.midnightblue), M7 (MM.brown) vs. M8 (MM.orange), M7 (MM.brown) vs. M10 (MM.pink), M9 (MM.blue) vs. M10 (MM.pink), M9 (MM.blue) vs. M12 (MM.turquoise), M11 (MM.darkorange) vs. M12 (MM.turquoise), M15 (MM.darkgrey) vs. M16 (MM.royalblue), M17 (MM.saddlebrown) vs. M18 (MM.salmon). However, no significant correlation was observed between other module combinations.

In order to understand the differences in functional features among the 18 modules, we also conducted GO enrichment and KEGG enrichment analyses (Appendix A). As expected, different modules did show differences in enriching various GO categories, although there were some common overlaps. For example, the enriched GO categories in M18 included “rhythm process”, “photomorphogenesis”, “regulation of response to red and far red light”, “circadian rhythm”, “response to light stimulus”, “long-day photoperiod”, and “response to hormone.”

In addition to the enriched GO terms in M18, we found that M1 module enriched in “regulation of shoot system development”, “carbohydrate catabolic process”, “regulation of flower development”, and “regulation of reproductive process.” Furthermore, representative GO categories enriched in other modules included “starch metabolic process (M2)”, “response to blue light (M3)”, “photosynthesis (M4)”, “pollen development (M7)”, “pollen wall assembly (M9)”, and “flower development (M10).”

In the enriched GO categories, known genes (or homologous genes in other species) that regulate plant bolting and flowering in response to photoperiod have been identified. Taking the M18 module as an example, key genes that regulate diurnal rhythm to promote flowering include *BraA10g000820.3C* (*LHY*), *BraA05g015000.3C* (*RVE2*), *MSTRG.37249* (*RVE1*), and *BraA05g036930.3C* (*RVE8*). Key genes that regulate physiological rhythms include *BraA02g006230.3C* (*COL1*) and *BraA03g035250.3C* (*LNK3*). Transcription factors that respond to light stimuli and promote photomorphogenesis include *BraA03g021010.3C* (*RCA*), *BraA03g038010.3C* (*HYH*), and *BraA04g022780.3C* (*BBX25*). Additionally, genes related to auxin response include *BraA10g015660.3C* (*XTH22*) and *BraA01g033190.3C* (*CYP705A22*), while genes related to abscisic acid response include *MSTRG.26548* (*AFP4*).

In the M2 module, key genes involved in photoperiodic flowering were identified, including *BraA03g023790.3C* (*SOC1*), *BraA05g026320.3C* (*JMJ30*), *MSTRG.13320* (*PAR2*), *BraA06g006390.3C* (*CCR1*), and the gibberellin biosynthesis key gene *BraA02g024710.3C* (*GA2OX1*). In the M12 module, key genes associated with photoperiodic flowering were *BraA03g061110.3C* (*GRDP2*), cell division hormone biosynthesis key gene *BraA07g030770.3C* (*IPT1*), *BraA09g051960.3C* (*ARR20*), key genes involved in abscisic acid response *BraA01g001770.3C* (*PLP3*) and *BraA10g031810.3C* (*GINT1*), key gene involved in jasmonic acid metabolism *BraA06g015190.3C* (*JMT*), key genes involved in carbohydrate transport *BraA01g022370.3C* (*MSSP3*) and *BraA01g035980.3C* (*NSP2*), and key genes involved in pollen development *BraA09g016530.3C* (*NAC025*) and *BraA05g013440.3C* (*BHLH91*).

In addition to these, other genes related to circadian rhythms were also identified, including *BraA08g015510.3C* (*CAT2*; M3), *BraA08g031180.3C* (*NPF6.3*; M6), *BraA03g058970.3C* (*GATA3*; M14), and *BraA02g037060.3C* (*LIR1*; M15).

### 2.12. Co-Expression Network of the DEGs Expressed in Tillers

To analyze the co-expression network of differentially expressed genes involved in the regulation of photoperiodic flowering, we constructed a weighted gene co-expression network. Out of 396 differentially expressed genes, we identified 61 differentially expressed genes that were connected to 5 gene categories (Figure 12, Appendix A). *BraA05g015000.3C-RVE2*, *BraA10g000820.3C-LHY*, *BraA03g038010.3C-HYH*, *BraA04g022780.3C-BBX25*, *BraA01g033190.3C-CYP705A22*, *MSTRG.37249-RVE1*, *BraA03g035250.3C-LNK3*, and *MSTRG.26548-AFP4* exhibited high connectivity in all 5 categories. *BraA02g006230.3C-COL1* showed low degree (28) but high connectivity (17.80).

Among the 10 flowering-related genes affected by light cycles, *BrRVE2* showed the highest connectivity, followed by *BrLHY* and *BrRVE1*. *BrRVE2* was connected to two light-responsive genes, two photosynthesis-related genes, three abscisic acid-related genes, and two auxin-related genes. *BrLHY* was related to *BrCOL1*, *BrBBX25*, *BrCYP705A22*, and *BrLNK3*. *BrRVE1* was related to *BrXTH22* and *BrELIP1* (Figure 12 and Figure 13).

## 3. Discussion

Plants not only use light energy as a source of energy for maintaining life, but also utilize light energy as a signaling factor to regulate and alter growth, development, and secondary metabolite synthesis, in order to maintain metabolic levels [31]. The influence of light energy mainly includes light intensity, light quality, and photoperiod, and these mechanisms are very complex, with various ways through which plants regulate plant growth and developmental processes [32]. Under a controlled environment, providing a relatively stable environment can minimize interference from biotic and abiotic factors [33]. This experiment aims to study the response of plants to intermittent light in a stable environment by using LED light tubes and controllers to regulate the light environment [34]. By controlling the light intensity and duration, we can alter the photoperiod environment in which the plants are exposed, and measured horticultural traits, morphological indicators, chlorophyll content, photosynthetic characteristic indicators, and transcriptomics [35]. Our research results reveal the impact of intermittent light on the early bolting and early flowering of ‘49 d’ flowering Chinese cabbage, and we also attempt to decipher its regulatory mechanism.

### 3.1. Intermittent Light Promotes Early Bolting of Flowering Chinese Cabbage

Changes in the light environment have a direct impact on plant growth and morphology [36]. In this experiment, we observed that intermittent light promoted the early bolting (50%, 6 days earlier), bud emergence (50%, 12 days earlier), and flowering (50%, 9 days earlier) of flowering Chinese cabbage. Additionally, previous studies showed that the growth rate of *Alocasia macrorrhiz* [37] and lettuce [38] remains unchanged and accelerates, respectively, under intermittent light. The conclusion of this experiment is consistent with previous research, indicating that intermittent light has different promoting effects on different species of plants, and different plants have different sensitivities to intermittent light. Furthermore, we found that the aboveground fresh weight, blade area, dry weight of blade, and quantification of leaves in treatments were lower than CK (Figure 2). However, the tiller’s height, dry weight of tillers, and tiller’s weight of the treatments were higher than the CK. Although this is contrary to the research results of lettuce [38] and cucumber [39], which show that intermittent light increases quantification of leaves and blade area of lettuce and improves seedling index of cucumber, we believe this may be due to the fact that the main edible part of flowering Chinese cabbage is the tillers, while the main edible parts of lettuce and cucumber are the leaves and fruits, respectively. However, the accelerated growth and development of different species are consistent. As for the decrease in the thickness of tillers, we believe this is determined by the characteristics of flowering Chinese cabbage itself. Although the measurement of the thickness of tillers was conducted at a fixed height in this experiment, during the late stage of flowering in flowering Chinese cabbage, the thickness of the tillers will reduce compared to the previous stage, due to the rapid elongation and growth of the tillers.

### 3.2. The Effect of Intermittent Light on the Photosynthetic Characteristics of Flowering Chinese Cabbage Leaves

Chlorophyll is an important photosynthetic pigment that plays a crucial role in the absorption, capture, and transfer of light energy. The content of chlorophyll and the activity of pigment proteins are closely related to light intensity [40]. In this experiment, it was observed that the contents of chlorophyll a, chlorophyll b, carotenoids, and total chlorophyll (a+b) contents, showed an increasing trend followed by a decrease, reaching the highest point in the S2 stage, and the chlorophyll a/b ratio in the treatments showed a decreasing trend. This is consistent with previous studies on the effects of intermittent light treatment on the chlorophyll content of Spinach [41] leaves. However, the difference in growth stages of flowering Chinese cabbage have different requirements for light intensity. We have found that white spots appear on the leaves of flowering Chinese cabbage during the seedling stage and bolting stage, which might be due to chloroplast damage. However, during the bud emergence stage and flowering stage, this light intensity is normal for flowering Chinese cabbage.

However, the net photosynthetic rate showed an increasing trend followed by a decrease, with the rate in CK being lower than treatments. These two contrasting results may be because intermittent light promoted the bolting of ‘49 d’ flowering Chinese cabbage, but the quantification of leaves and blade area of treatments were lower than CK. Therefore, the SPAD value of the leaves in treatments was relatively lower, indicating that intermittent light increased the photosynthetic rate of the leaves, mainly manifested in promoting the elongation and development of the tillers.

Chlorophyll fluorescence parameters are key indicators for determining the level of stress or damage to plant photosynthetic apparatus [42]. We found that the *Fv/Fm* values varied between 0.75 and 0.81 throughout the entire growth cycle, which may be related to the high light intensity experienced during the seedling stage [43]. In addition, there was an increasing trend in the changes of *Fv/Fm*, with higher values observed in treatment compared to CK after the S3 stage. The qP and NPQ values in the treatments were higher than CK in S1, S2, and S3 stages, while the ETR values in the treatments were consistently lower than CK, although the difference gradually decreased. This suggests that the stress damage caused by high light intensity during the seedling stage is reversible, and the PSII electron transfer activity is stronger under intermittent light treatment [44,45], which can accelerate the recovery time for damage [46].

### 3.3. The Effect of Intermittent Light on the Nutritional Indicators and Antioxidant Capacity of Flowering Chinese Cabbage

Soluble sugars, soluble proteins, vitamin C, and nitrate content are important indicators for evaluating the nutritional quality and taste of leafy vegetables [47]. Nitrate tends to accumulate in vegetables, especially leafy vegetables, and poses potential health risks [48]. In this experiment, we found that the content of soluble sugars, soluble proteins, and vitamin C in tillers and leaves of treatments was higher than the CK. Among them, the change in soluble sugar content in tillers was the most significant, which may be due to the promotion of tillers development through intermittent light. The nitrate content in tillers and leaves of the treatments was relatively lower than the CK, and the reduction increased with the progress of the biological process. This indicates that treatments accelerated the growth of flowering Chinese cabbage, promoted the development of tillers at the same stage, and improved the taste and quality.

The content of the phenolics flavonoids, DPPH free radical scavenging rate and FRAP iron ion reduction capacity can usually reflect the total antioxidant capacity in vegetables [49]. In this experiment, we found that the content of phenolics flavonoids in tillers and leaves of treatments increased, indicating that intermittent light improved the antioxidant capacity of flowering Chinese cabbage at the same stage, promoted the synthesis of secondary metabolites such as flavonoids, and allocated more energy to the reproductive growth of flowering Chinese cabbage. As for DPPH and FRAP, the results showed differences. The values of both the treatments in tillers and leaves were higher than the CK. In addition, the DPPH in the leaves remained relatively stable, only decreasing in the S4 stage, while the FRAP in the tillers started to decrease in the S2 stage, but the change was similar to the CK in next stage, possibly due to the promotion of the development process via intermittent light. Finally, the different performance and trend of the treatments in the S4 stage may be mainly due to the flowering performance.

### 3.4. The Effect of Intermittent Light on the Mineral Content of Flowering Chinese Cabbage

The accumulation of sodium (Na) does not have an effect on the accumulation of potassium (K), magnesium (Mg), and calcium (Ca) [50]. From the perspective of mineral nutrition, intermittent light can promote the accumulation of nitrogen (N), K, Ca, sulfur (S), and zinc (Zn) elements in the tillers, but has no effect on phosphorus (P) and Mg elements. Since N, Mg, P, K, and S are involved in the synthesis of certain plant hormones, the biosynthesis of chlorophyll, the catalysis of key enzymes involved in photosynthesis, and the transportation of carbohydrates and electrons [51] the increased content of N, K, and S caused via intermittent light may contribute to the increase in the content of photosynthetic pigments, promote photosynthesis, help produce carbohydrates, and thereby promote the growth of tillers. However, the slight decrease in Mg and P elements may be due to the limited number of leaves. In addition, light has a significant effect on the accumulation of Ca, indicating that Ca may play an important role in transmitting light signals and regulating signal transduction in plant growth and development.

### 3.5. Investigation of the Molecular Mechanism of Intermittent Light on the Early Bolting of Flowering Chinese Cabbage

In this study, we focused on exploring the impact of intermittent light on the key genes related to bolting in flowering Chinese cabbage. Through the observation of the sample clustering map, we found a high correlation between the S1-1s and S2-CK, which is consistent with the phenotype. We analyzed the intersection of these three combinations and the intersection of the treatments and the CK at the same time point and obtained a total of 396 DEGs. Interestingly, among the selected key genes, the highest proportion was related to circadian rhythm genes (44.12%), and growth and development-related genes had the highest proportion in regulating pollen and anther development (71.43%). The findings indicating that intermittent light may disrupt the original light cycle by altering the circadian rhythm of flowering Chinese cabbage, promoting the bolting and internal development of pollen and anthers, ultimately leading to early bud formation and flowering.

According to our observations, we found that bolting of the treatments in the S2 stage. In order to further investigate, we selected 15 genes related to circadian rhythm for quantitative measurement. The experimental results showed that the expression levels of *BrRVE8*, *BrRVE1*, *BrCOL1*, *BrLHY*, and *BrRVE2* in the S1-1s and S2-CK were similar. Based on these results, we speculate that these genes may respond to intermittent light and may be related to promoting bolting in flowering Chinese cabbage.

The research on the biological clock has been extensively studied in *Arabidopsis thaliana*. LHY (late elongated hypocoty1), CCA1 (circadian clock associated 1), and the pseudo-response regulators (PRRs) form the core of the biological clock [52]. They can process signals and transmit them to *GI* (*gigantea*), thereby regulating downstream flowering genes and controlling the flowering process in *Arabidopsis thaliana* [53,54,55]. There have been experiments proving that *LHY* overexpression in plants results in a late-flowering phenotype [56]. The flowering time of *lhy* in long-day conditions is the same as the wild-type, but it advances in short-day environments [57]. Our research results show that *BrLHY* in the treatments decreases during stage S2, while there is little change in the CK. This indirectly indicates that intermittent light promotes the early bolting of flowering Chinese cabbage by disrupting the intrinsic biological clock rhythm. However, so far, we have not obtained information about *BrLHY* overexpression, so further research on the function of *BrLHY* is needed.

*RVEs* (*REVEILLEs*) are a class of MYB-type transcription factors that contain a single MYB DNA-binding domain [58]. In the core oscillator of the plant circadian clock, transcription factors such as *RVE4*, *RVE6*, and *RVE8* are key components along with CCA1 and LHY, while *RVE3* and *RVE5* have a smaller impact on clock function [59]. Additionally, *RVE1* enhances the effect of auxin by positively regulating the expression of the auxin biosynthesis gene *YUCCA8* (*YUC8*) [60]. In our study, we found that the expression patterns of *BrRVE1*, *BrRVE2*, and *BrRVE8* are similar to that of *BrLHY*, suggesting their possible involvement in the early bolting phenomenon induced via intermittent light. However, it is currently unclear whether the overexpression of *BrRVE1*, *BrRVE2*, and *BrRVE8* can promote flowering, and further functional studies are needed.

To further investigate the interactions between genes involved in intermittent light-enhanced flowering response, we established a co-expression network and classified the genes into five categories [61,62]. Interestingly, we found high connectivity among *BrRVE2*, *BrLHY*, *BrHYH*, *BrBBX25*, *BrCYP705A22*, *BrRVE1*, *BrLNK3*, and *BrAFP4*. Even though *BrCOL1* had low abundance, it also exhibited high connectivity. Among the 10 flowering-related genes affected by 10 photoperiods, *BrRVE2* showed the highest connectivity, followed by *BrLHY* and *BrRVE1*. *BrRVE2* was connected to two light-responsive genes, two photosynthesis-related genes, three abscisic acid-related genes, and two auxin-related genes.

## 4. Materials and Methods

### 4.1. Plant Materials and Experimental Procedures

The experiment was conducted in South China Agricultural University (latitude 23°9′40″ N, longitude 113°21′18″ E). The flowering Chinese cabbage varieties were purchased from Guangdong Kenong Vegetable Seed Industry Co., Ltd (Guangzhou, Guangdong Province, China).

The seedlings of flowering Chinese cabbage are planted in a small flower pot with an outer diameter of 6 cm. The potting soil is mixed with soil, vermiculite, and perlite in a volume ratio of 3:1:1. The abundant elements in the cultivation nutrient solution adopt a 1/2 Hoagland solution, while the trace elements adopt a general formula, with the pH adjusted to 6–6.5. The seeds are soaked in a 5% sodium hypochlorite solution for 10 min for disinfection, and then rinsed with deionized water 3–4 times. The rinsed seeds are evenly spread on a culture dish covered with gauze, and the appropriate moisture is maintained. The culture dish is placed in the dark for overnight cultivation until the seeds turn white and the germination rate reaches or exceeds 80%, and then the seeds are sown in the small flower pot.

The laboratory is powered on all day, and the lights are controlled using a controller. The lights are set to turn on at 8:00 a.m., defined as zeitgeber time 0 (ZT0), and turn off at 20:00 p.m., defined as zeitgeber time 12 (ZT12). During the seedling stage before the cotyledons of the cabbage seedlings flatten, they receive 12 h of light and 12 h of dark treatment.

In order to study the effects of intermittent light on bolting and flowering of flowering Chinese cabbage, the cabbage seedlings start to receive intermittent light treatment after growing to the stage of three leaves. This treatment includes the CK group and a 1s group. In the 1s group, the lights are set to turn on at 8:00 a.m., defined as zeitgeber time 0 (ZT0), and turn off at 8:01 a.m., defined as zeitgeber time 001 (ZT001), with a 24 h cycle repeated.

### 4.2. Changes in Growth Parameters and Determination of Horticultural Traits

Record the occurrence of bolting, bud emergence, and flowering in the CK and treatments throughout the entire growth cycle, and calculate the bolting rate, bud emergence rate, and flowering rate, based on the total number of potted plants, by dividing the number of plants with the corresponding phenotype by the total number of plants. Subsequently, divide the entire experiment into five stages based on the time at which the bolting rate, bud emergence rate, and flowering rate reach or exceed 50%, namely S0 (cotyledon expansion stage, 0 d), S1 (two-leaf stage in the treatment and control groups, 11 d), S2 (treatments with bolting rate over 50% and control group without bolting, 17 d), S3 (treatments with budding rate over 50% and control group with bolting, 26 d), and S4 (treatments with flowering rate over 50% and control group with budding, 32 d). Take photos of representative plants during the aforementioned stages.

In addition, it is also necessary to conduct statistical and measurement analysis on the aboveground fresh weight (excluding cotyledons), blade area, dry weight of blade, quantification of leaves, thick of tillers, tillers height (excluding flower buds), dry weight of tillers, and tillers weight for both groups.

### 4.3. Chlorophyll Content, Photosynthetic Characteristic Indicators, and Chlorophyll Fluorescence Characteristics Determination

The chlorophyll a content, chlorophyll b content, and carotenoid content in the CK and treatments leaf blades at stages S1, S2, S3, and S4 were measured using spectrophotometric analysis [63]. Additionally, the total chlorophyll (a+b) content and chlorophyll a/b ratio were calculated.

We used the TARGAS-1 portable photosynthesis system (brand: PP System) to track and measure the net photosynthetic rate (Pn) of the leaf blades in four directions, namely east, west, south, and north, for the mustard seedlings. In the S1 stage, we only measured the leaf blades in the east and west directions.

Additionally, we also used the SPAD-502PLUS chlorophyll meter to measure the SPAD values of each group of leaf blades.

In addition, using the chlorophyll fluorescence imaging system (IMAGINE PAM) in the public experimental laboratory of the College of Horticulture, we conducted measurements [64].

### 4.4. Nutritional Quality and Physiological Indicators Determination

We measured the nitrate content, vitamin C content, soluble sugar content, and soluble protein content of leaf blades and tillers in the CK and treatments at four stages, S1, S2, S3, and S4. The measurement of soluble sugar content was conducted using the anthrone colorimetric method [65], the measurement of soluble protein content was conducted using the Coomassie brilliant blue method [66], the measurement of vitamin C content was conducted using the molybdenum blue colorimetric method [67], and the measurement of nitrate content was conducted using the sulfanilic acid-salicylic acid colorimetric method [68].

In addition, we also measured the total phenolic content and total flavonoid content of antioxidant substances, as well as the DPPH and FRAP antioxidant capacities. The total phenolic content was determined using the Folin–Ciocalteu method [69], and the total flavonoid content was determined using the aluminum nitrate method [70]. The DPPH free radical scavenging activity [71] and FRAP antioxidant capacity [72] were determined according to previous methods.

### 4.5. Mineral Element Determination

Samples of oven-dried flowering Chinese cabbage were weighed, ground to powder, and stored to measure mineral element contents. Total nitrogen (N), phosphorus (P), potassium (K), calcium (Ca), magnesium (Mg), sulfur (S), zinc (Zn), and iron (Fe) was measured by using an atomic absorption spectrophotometry method [73].

### 4.6. RNA Extraction, Sequencing, and De Novo Sssembly

Samples from five stages of treated and control groups were collected and frozen in liquid nitrogen, then stored at -80℃ for RNA extraction and library construction. Total RNA extraction was performed using the Plant Total RNA Isolation kit (Huayueyang, Beijing, China). The integrity of the extracted RNA was assessed using a 1.2% denaturing agarose gel and the Agilent Bioanalyzer Model 2100 (Agilent Technologies, Santa Clara, CA, USA). High-purity RNA samples were used for cDNA library construction. The quality and quantity of the libraries were validated using the Agilent Bioanalyzer Model 2100 and real-time RT-PCR, respectively. Subsequently, MetWare conducted sequencing of 27 purified cDNA libraries using the Illumina HiSeqTM 2500 platform (Illumina, San Diego, CA, USA).

The original fastq format was processed using an internal Perl script. In this step, clean reads were obtained by removing adapter sequences, reads containing 10% N bases, and/or reads with a base quality ratio >50% for bases with Q ≤ 20. The copyright of the Perl script belongs to Metware (http://www.metware.cn/ accessed on 20 February 2024). Additionally, Q20, Q30, and GC content were calculated. Ribosomal reads were removed using Bowtie [74]. Then, cleaned reads were mapped to the flowering Chinese cabbage genome (http://39.100.233.196:82/download_genome/Brassica_Genome_data/Brara_Chiifu_V3.0/ (accessed on 20 February 2024).

Based on the alignment results of HISAT2, we reconstructed transcripts using Stringtie [75] and calculated the expression levels of all genes in each sample using RSEM [76]. The read counts and fragments per kilobase of transcript per million mapped reads (FPKM) of each gene were calculated using eXpress (https://pachterlab.github.io/eXpress/index.html (accessed on 24 April 2023). Differentially expressed genes (DEGs) were determined using OmicShare tools (http://www.omicshare.com/tools/Home/Soft/diffanalysis (accessed on 24 April 2023)). The R package is edgeR (http://www.bioconductor.org/packages/release/bioc/html/edgeR.html (accessed on 24 April 2023, and significant DEGs were restricted with false discovery rate (FDR) ≤ 0.05 and the absolute value of fold-change ≥ 2.

### 4.7. Transcriptome Data Analysis

Transcripts assembled using Trinity were the number of unigenes. The unigenes were used for BLASTX alignment and annotation against seven public databases (KEGG, NR, SwissProt, Trembl, KOG, GO, and Pfam). The expression of unigenes was calculated based on their FPKM (fragments per kilobase of transcript per million mapped fragments) values with K-means method. The identification of differentially expressed genes (DEGs) was performed using DESeq2 [77]. In this study, a false discovery rate (FDR) < 0.05 and an absolute value of |log2 FC)| > log2(2)were used as the threshold to determine significant DEGs.

To identify the putative biological functions and pathways for the DEGs (differentially expressed genes), GO (gene ontology) and KEGG (Kyoto encyclopedia of genes and genomes) enrichment analyses were conducted using OmicShare Tools. In addition, GO and KEGG pathways with a q-value < 0.05 were significantly enriched in DEGs.

### 4.8. Validation of RNA-Seq Data Using QRT-PCR

The first-strand cDNA was synthesized using the Reverse Transcriptase M-MLV (RNase H-) system (Takara, Dalian, China) from 1 μg of extracted RNA. The quantitative real-time polymerase chain reaction (qRT-PCR) primers, F1/R1 (Supplementary File 8: Appendix A), were designed using Primer 6.0 software (Premier Biosoft, Palo Alto, USA) and synthesized by Sangon Ltd. (Shanghai, China). Flowering Chinese cabbage homologous reference gene GADPH was used as the reference gene (Appendix A). qRT-PCR was performed on a Light Cycler480 real-time PCR machine (Roche, Basel, Switzerland) [78]. The transcriptional quantification of genes was relative to the reference gene and calculated using the 2−^ΔΔCT^ method [79]. The analysis was conducted with three biological replicates and three technical replicates.

### 4.9. Construction of Co-Expression Networks for Genes

I chose the Weighted Gene Co-Expression Network Analysis (WGCNA, v1.48) package in R to obtain the weight values of selected genes. I used Cytoscape (version 3.2.1) to analyze the connectivity between genes and construct the network. A threshold of 0.12 was set for weight values, and the color of the connecting lines (edges) ranged from light pink to brown, corresponding to the weight values, with darker colors indicating higher values. The size of the nodes corresponds to the connectivity obtained from Cytoscape’s NetworkAnalyzer analysis, including degree (number of edges) and weight values.

### 4.10. Statistical Analysis

Data were analyzed using SPSS (version 19.0; IBM Corp., Armonk, NY, USA) using Student’s *t*-test and Duncan’s multiple range test. The expression of candidate genes is presented in the form of Heatmaps or line graphs using the Tbtools package. The graphs were generated using GraphPad Prism (version 8.0.2).

### 4.11. Accession Numbers

The transcriptome data were submitted into NCBI’s SRA (sequence Read Archive) database using accession no. SRR28051825, SRR28051824, SRR28051823, SRR28051822, SRR28051821, SRR28051820, SRR28051819, SRR28051818, SRR28051817, SRR28051816, SRR28051815, SRR28051814, SRR28051813, SRR28051812, SRR28051811, SRR28051810, SRR28051809, SRR28051808, SRR28051807, SRR28051806, SRR28051805, SRR28051804, SRR28051803, SRR28051802, SRR28051801, SRR28051800, SRR28051799.

## 5. Conclusions

Based on the above, by setting intermittent light in the facility environment, we can promote the bolting, bud emergence, and flowering process of flowering Chinese cabbage earlier, thereby improving the photosynthetic rate and characteristics. At the same time, intermittent light also inhibits the occurrence of new leaves, promotes the growth and development of the tillers, and consequently promotes the maturity of flowering Chinese cabbage, shortening the growth cycle, improving its nutritional quality and antioxidant activity, and accumulating more mineral elements. In our study, through transcriptome analysis, we have identified three genes, *BrRVE2*, *BrLHY*, and *BrRVE1*, which are most likely involved in the regulation of early bolting of flowering Chinese cabbage by intermittent light. Next, we need to further investigate the functions of these genes.

## Figures and Tables

**Figure 1 plants-13-00866-f001:**
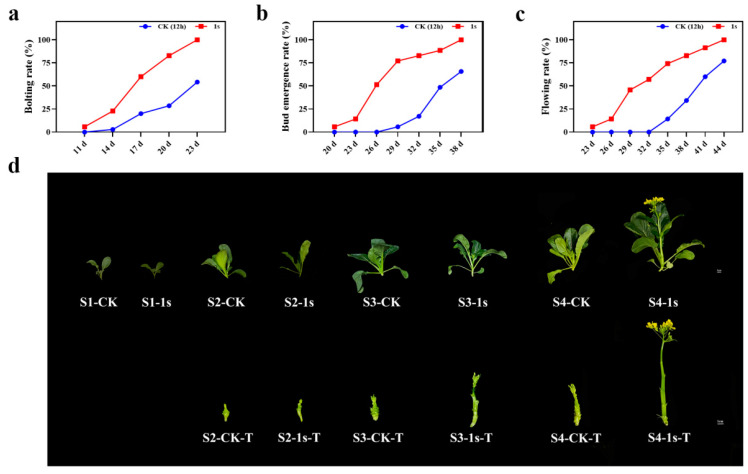
The influence of intermittent light on the statistical data and morphological change in the growth and development of ‘49 d’ flowering Chinese cabbage. (**a**) The effect of intermittent light on the bolting rate of flowering Chinese cabbage. (**b**) The effect of intermittent light on the bud emergence rate. (**c**) The impact of intermittent light on the flowering rate. The scale in Figure (**d**) is 1 cm, displaying a comparative graph between the CK and treatments for the all stages. The bottom panel in the figure shows a comparative graph of the changes in tillers without leaves during the S2, S3, and S4 stages, T: tiller.

**Figure 2 plants-13-00866-f002:**
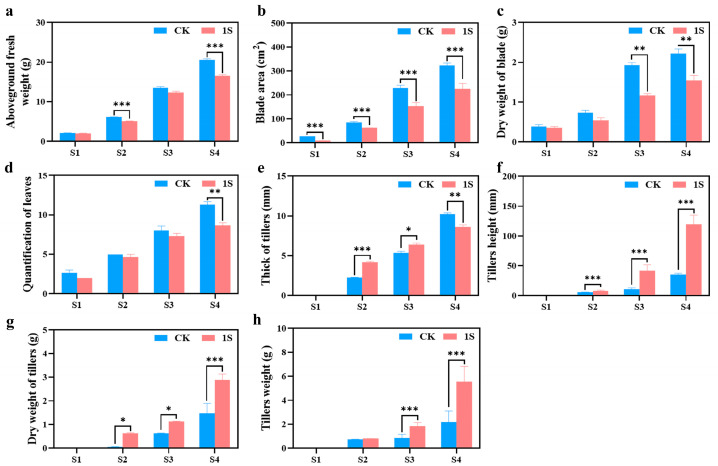
The effects of intermittent light treatment; the morphological indicators of ‘49 d’ flowering Chinese cabbage have changed. (**a**) Aboveground fresh weight. (**b**) Blade area. (**c**) Dry weight of blade. (**d**) Quantification of leaves. (**e**) Thickness of tillers. (**f**) Tiller’s height. (**g**) Dry weight of tillers. (**h**) Tiller’s weight. Independent samples *t*-test, * indicates a significant difference between treatment and control at the *p* ≤ 0.05 level (n = 9), ** indicates a significant difference at the *p* ≤ 0.01 level, and *** indicates a significant difference at the *p* ≤ 0.001 level.

**Figure 3 plants-13-00866-f003:**
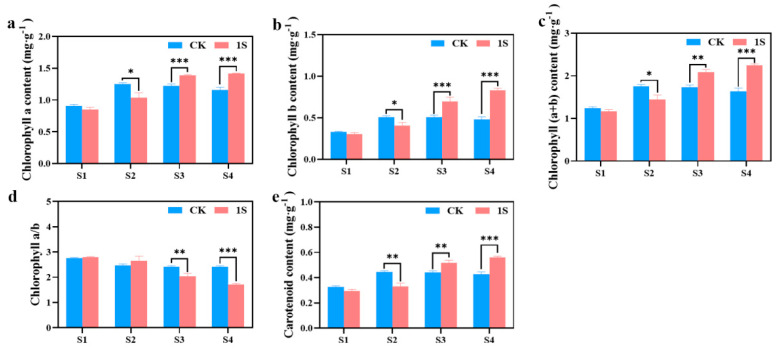
The effects of intermittent light treatment on chlorophyll content of ‘49 d’ flowering Chinese cabbage were observed. (**a**) Chlorophyll a content. (**b**) Chlorophyll b content. (**c**) Total chlorophyll (a+b) content. (**d**) Chlorophyll a/b. (**e**) Carotenoid content in flowering Chinese cabbage. Independent samples *t*-test, * indicates a significant difference between treatment and control at the *p* ≤ 0.05 level (n = 9), ** indicates a significant difference at the *p* ≤ 0.01 level, and *** indicates a significant difference at the *p* ≤ 0.001 level.

**Figure 4 plants-13-00866-f004:**
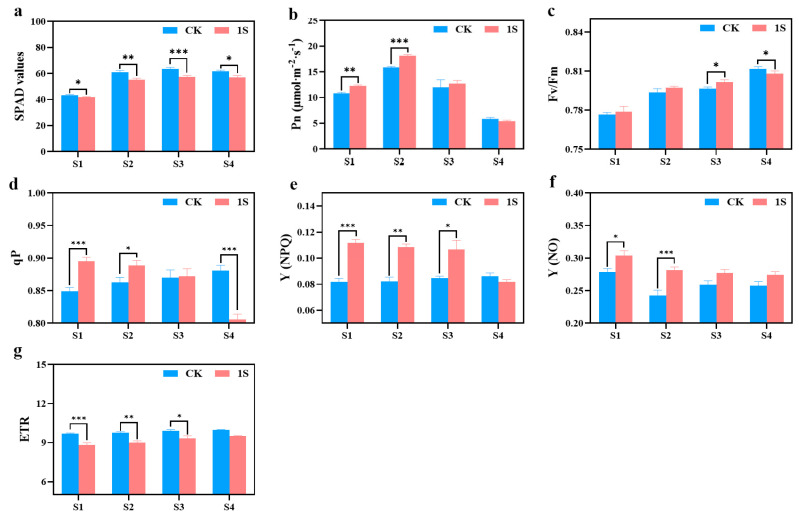
The effects of intermittent light treatment on photosynthetic characteristics, and chlorophyll fluorescence characteristics of ‘49 d’ flowering Chinese cabbage were observed. (**a**) SPAD values. (**b**) Photosynthetic rate. (**c**) Fv/Fm values. (**d**) qP values. (**e**) (NPQ) values. (**f**) (NO) values. (**g**) ETR values in flowering Chinese cabbage. Independent samples *t*-test, * indicates a significant difference between treatment and control at the *p* ≤ 0.05 level (n = 9), ** indicates a significant difference at the *p* ≤ 0.01 level, and *** indicates a significant difference at the *p* ≤ 0.001 level.

**Figure 5 plants-13-00866-f005:**
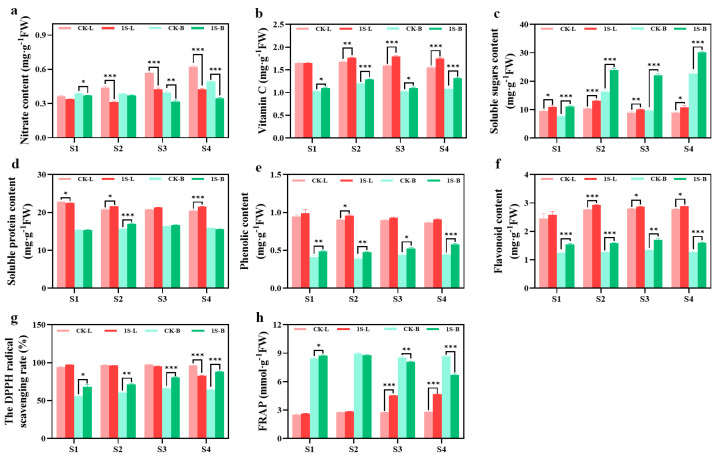
The effects of intermittent light on the nutrient and antioxidant content, and antioxidant capacity of ‘49 d’ flowering Chinese cabbage leaves and tillers. (**a**) Nitrate content. (**b**) Vitamin C content. (**c**) Soluble sugar content. (**d**) Soluble protein content. (**e**) Phenolic content. (**f**) Flavonoid content. (**g**) DPPH values. (**h**) FRAP values in flowering Chinese cabbage. Independent samples *t*-test, * indicates a significant difference between treatment and control at the *p* ≤ 0.05 level (n = 9), ** indicates a significant difference at the *p* ≤ 0.01 level, and *** indicates a significant difference at the *p* ≤ 0.001 level.

**Figure 6 plants-13-00866-f006:**
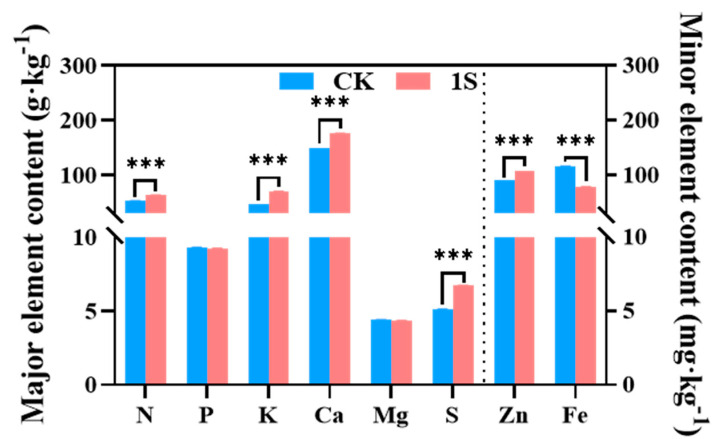
The effects of intermittent light on the major and minor elements of ‘49 d’ flowering Chinese cabbage. *** indicates a significant difference at the *p* ≤ 0.001 level.

**Figure 7 plants-13-00866-f007:**
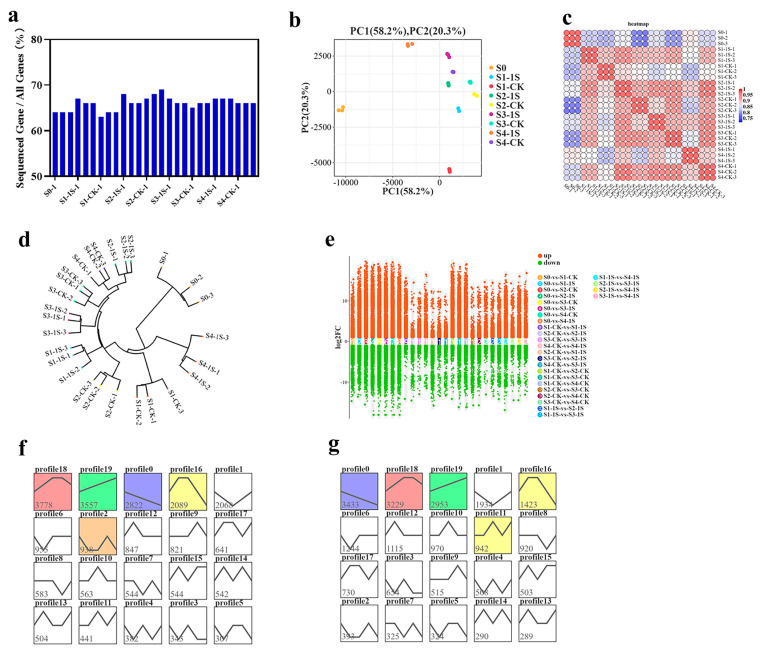
Gene expression analysis results of ‘49 d’ flowering Chinese cabbage under intermittent light treatment. (**a**) Ratio of expressed genes to total identified genes in each sample. (**b**) Principal component analysis of the samples. (**c**) Correlation statistics plot among the samples. (**d**) Cluster analysis plot of the samples. (**e**) Scatter plot of differentially expressed genes (DEGs) with multiple-group differences. (**f**) Comparative results in the major gene expression patterns, with colored contours indicating their statistical significance (*p* ≤ 0.05). The same below. (**g**) Comparative results between S0 and CKs in the major gene expression patterns.

**Figure 8 plants-13-00866-f008:**
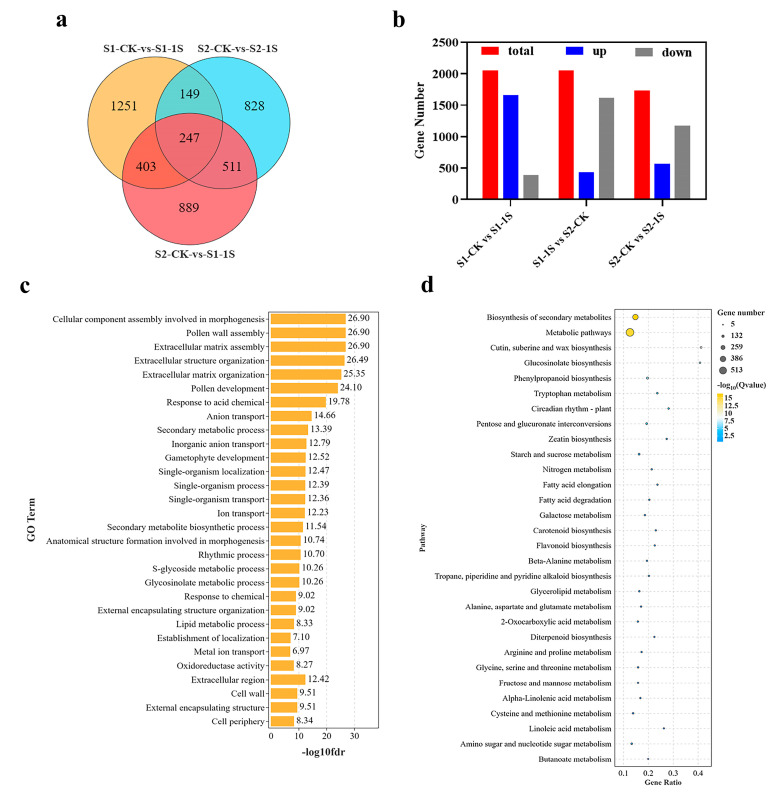
Differential gene analysis promoting early bolting of ‘49 d’ flowering Chinese cabbage under intermittent light treatment. (**a**) Venn diagram showing the number of DEGs among the 3 comparison groups. (**b**) Proportion of upregulated and downregulated genes among the 3 DEG groups. (**c**) Functional enrichment of the 3 DEG groups. (**d**) Enrichment of the 3 DEG groups in metabolic pathways.

**Figure 9 plants-13-00866-f009:**
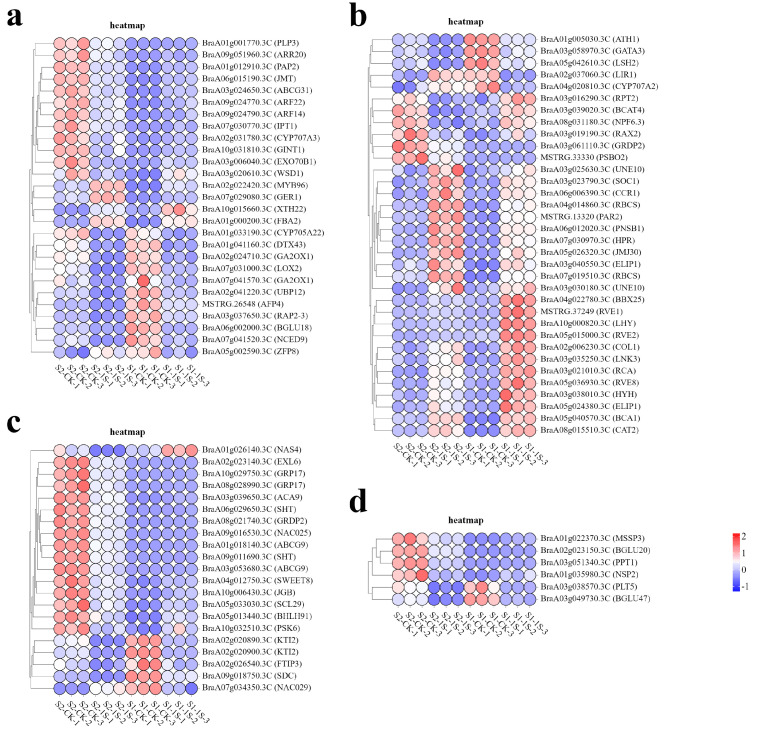
Expression patterns of DEGs in response to intermittent light conditions. (**a**) Heatmap of DEGs related to hormones. (**b**) Heatmap of DEGs related to light. (**c**) Heatmap of DEGs related to development. (**d**) Heatmap of DEGs related to carbohydrates.

**Figure 10 plants-13-00866-f010:**
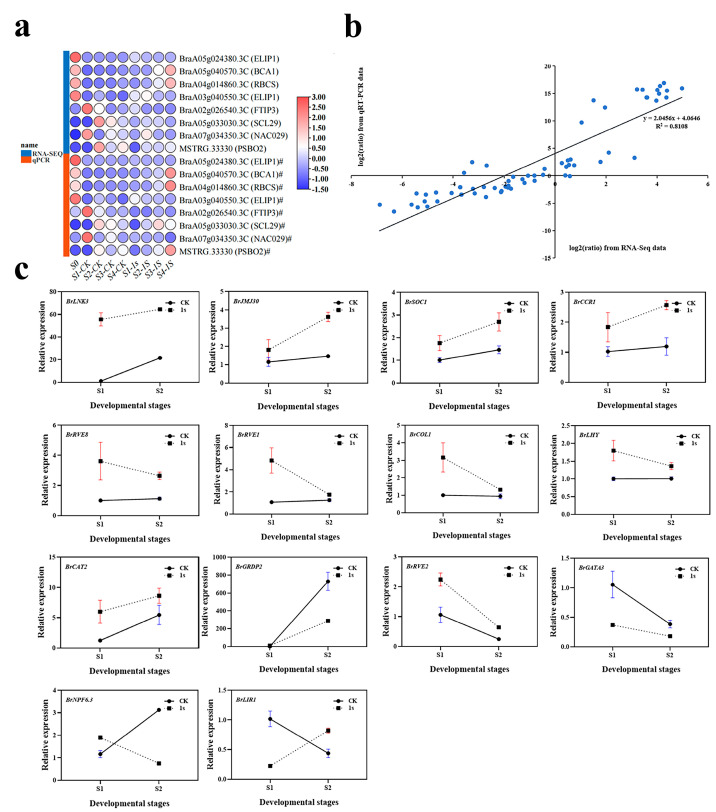
(**a**) Heatmap of test gene expression levels shown using quantitative real-time RT-PCR (qRT-PCR) and RNA sequencing (RNA-Seq). (**b**) Correlation between qRT-PCR and RNA-Seq for test genes. (**c**) Expression profile of candidate genes in the early bolting stage of ‘49 d’ flowering Chinese cabbage induced via intermittent light stimulation. Relative transcription was calculated through qRT-PCR using the 2^−ΔΔCT^ method with actin as a reference. Data are means of three replicates.

**Figure 11 plants-13-00866-f011:**
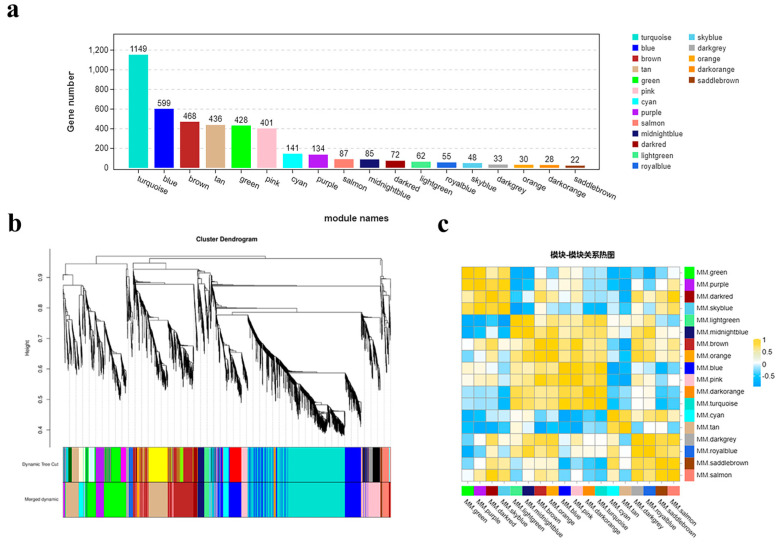
Co-expression modules involved in early bolting of flowering Chinese cabbage. (**a**) Eighteen modules identified through WGCNA and the number of genes in each module. (**b**) Hierarchical clustering of the eighteen modules identified through WGCNA. (**c**) Heatmap depicting the relationships between different co-expression modules.

**Figure 12 plants-13-00866-f012:**
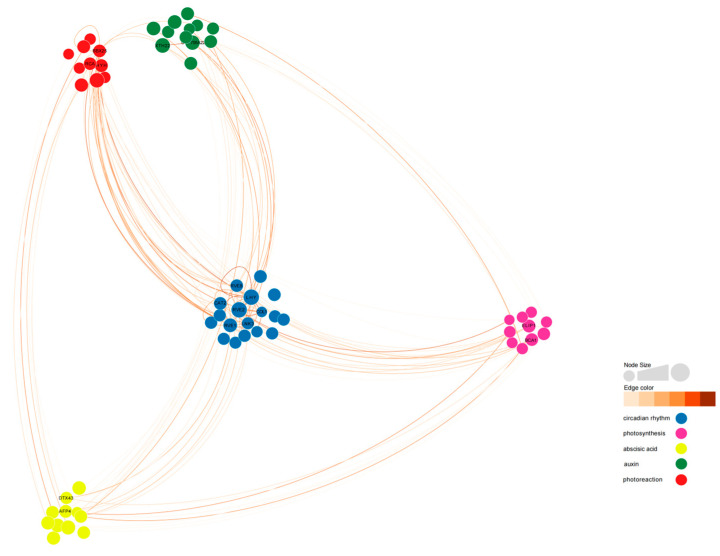
Co-expression network related to circadian rhythm in ‘49 d’ flowering Chinese cabbage. Ninety-three DEGs were selected among the 396 DEGs by using the weighted gene co-expression network analysis (WGCNA) R package, and a weight value >0.12 was subjected as the threshold value. Detailed information on the nodes and edges is shown in Appendix A.

**Figure 13 plants-13-00866-f013:**
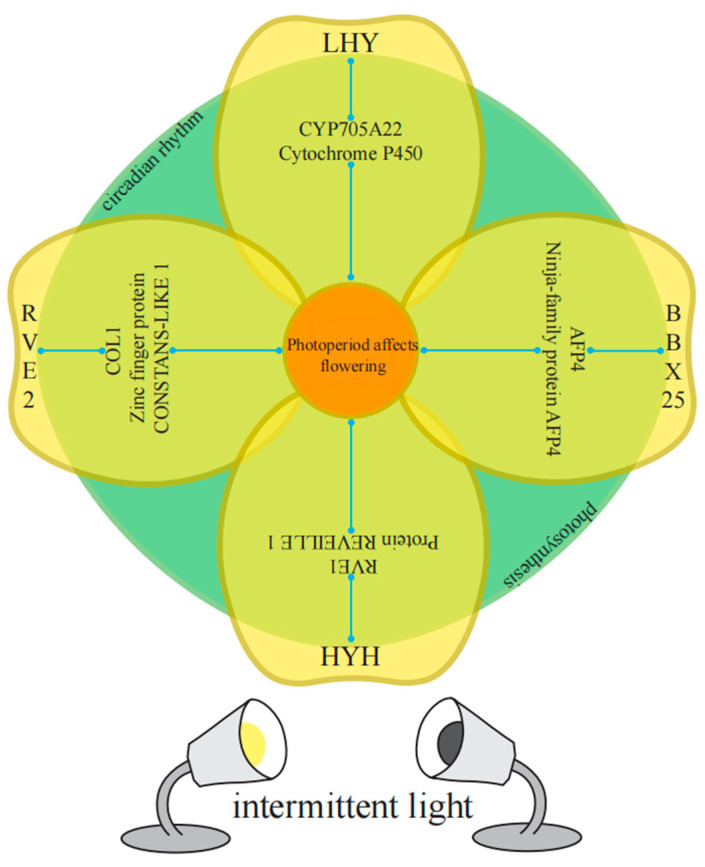
Co-expression network of flowering-related genes reveals detailed information about four identified flowering-related genes in the diurnal rhythm-related co-expression network. *BrLHY* is associated with *BrCYP705A22*, *BrRVE2* is connected to the *C2H2* zinc finger transcription factor gene *BrCOL1*, *BrHYH* is related to *BrRVE1*, and *BrBBX25* is negatively correlated with the abscisic acid-related gene *BrAFP4*.

## Data Availability

All data, models, or code generated or used during the study are available from the corresponding author on request.

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
