# Peer review of "Transcriptome Analysis of Intermittent Light Induced Early Bolting in Flowering Chinese Cabbage"

_plants, 2024, doi:10.3390/plants13060866_

Round 1
Reviewer 1 Report
Comments and Suggestions for Authors
The authors characterized the growth and physiological traits under intermittent light conditions in flowering Chinese cabbage. Many traits have been investigated, which provides useful information. First of all, I would like the authors to clarify whether intermittent light is effective for yield or not. Regarding RNA-seq analysis, I have some concerns as follows.
Why did you use an inflorescence tissue sample?
Is it an effective stage to study floral induction?
Is there already inflorescence tissue at S0 stage?
Why don't you show zeitgeber time when you harvest the sample? Because the authors are discussing about the clock genes.
With so many genes changing in expression, it is difficult to capture what causes early flowering. Therefore, the authors should indicate the expression pattern of the floral integrator first.
L110 I cannot understand which stage S1-S4 represents. The method is in the back , so it is easier to understand if it is written in the results part.
Author Response
请参阅附件。

Reviewer 2 Report
Comments and Suggestions for Authors
The regulation of flowering time is a very important issue, especially in the context of crop plants.
In the publication presented here, the authors analyzed the effect of intermittent lighting on flowering time, gene expression of physiological parameters related to photosynthesis and accumulation of nutrient compounds.
Overall, the study was conducted properly, the data was analyzed using appropriate methods and the results obtained are well documented. The authors showed that, as in other plants, in Chinese cabbage intermittent light promotes early flowering. They also indicated genes, including those related to circadian rhythm, whose expression is altered under the conditions tested.
However, before accepting publication in print, it is necessary to provide some important information. I also suggest making some minor changes, described below.
Major comments:
In conclusions, the authors state that the observed changes are due to the regulation of flowering by MYB-like transcription factors (RVE1, RVE2 and LHY). In my opinion, this is not clear from the data presented. In the Discussion (lines 613 to 619) it is stated that 'tThere have been experiments proving that LHY overexpression in plants results in a late-flowering pheno-type [56]. The flowering time of lhy in long-day conditions is the same as the wild type, but it advances in short-day environments [57]. Our research results show that BrLHY in the treatments is decreasing during stage S2, while there is little change in the CK. This indirectly indicates that intermittent light promotes the early bolting of flowering Chinese cabbage by disrupting the intrinsic biological clock rhythm’ whereas in Fig. 10, the expression of LHY decreases under treatment, but its relative expression is still higher than that of the control samples.
In addition, in Fig. 9 (b) (heat map of DEGs related to light) there are no LHY and RVE1 genes, does this mean that their expression did not change significantly? Can you explain this?
The description of the analysis of RNAseq results is not clear. In paragraph 4.6 there is information about mapping RNAseq results to a reference genome and in the next paragraph there is a description of transcriptome assembly, annotation and gene expression analysis. It is not clear for what purpose the reads were mapped to the reference genome, it seems to DEGs were called based on transcriptome assembly. In the material and methods, the database and ID of the reference genome used and the database and ID of the RNAseq results should be provided. All produced data, such as RNAseq, should be should be publicly available.
Minor comments:
Assign the time (OS X) in Fig. 1 a, b and c to the four stages of development described later in the manuscript.
I propose the generalization of the part of the abstract describing the results of the analysis of parameters related to photosynthesis and nutrient content.
Citation should be completed in the Materials and Methods, especially for programs/tools used to analyze RNAseq data.
Round 2
Reviewer 2 Report
Comments and Suggestions for Authors
Thank you for the clarification and improvement of the manuscript. Unfortunately, two important elements have still not been completed.
Line 737 - please add ID and the database for the 'flowering Chinese cabbage genome'.
I know that upload of raw RNAseq data (not only assembled transcripts) will take some time, but it shoud be done before submiting the manuscript. Access to the results on which the analyses are based is essential for acceptance of the publication for publication. As a reviewer, I need to be able to check that the results are correct.
Round 3
Reviewer 2 Report
Comments and Suggestions for Authors
I still do not see the possibility to repeat the experiment due to the lack of information about the reference genome used, the link provided https://brassicadb.cn/#/ is a reference to a database from which various genomes of different species of the genus Brassicaceae can be downloaded. Was the assembly published by Bayer, P. E. (2022). Brassica napus, rapa, oleracea pangenome assemblies and annotations (1.0) [Data set]. Zenodo. https://doi.org/10.5281/zenodo.7396989 used for this analysis.If so, it should be properly cited; if not, provide another source.
For raw reads, please provide the bioproject number and SRR numbers for the reads. The SUB number is irrelevant.
Author Response
Comments and Suggestions for Authors
Thank you for the clarification and improvement of the manuscript. Unfortunately, two important elements have still not been completed.
Reviewer Suggestion; I still do not see the possibility to repeat the experiment due to the lack of information about the reference genome used, the link provided https://brassicadb.cn/#/ is a reference to a database from which various genomes of different species of the genus Brassicaceae can be downloaded. Was the assembly published by Bayer, P. E. (2022). Brassica napus, rapa, oleracea pangenome assemblies and annotations (1.0) [Data set]. Zenodo. https://doi.org/10.5281/zenodo.7396989 used for this analysis.If so, it should be properly cited; if not, provide another source..
Author Response; thanks for your kind suggestion and i am sorry for my mistake. We have added the information about the reference genome used
The original fastq format was processed using an internal Perl script. In this step, clean reads were obtained by removing adapter sequences, reads containing > 10% N bases, and/or reads with a base quality ratio > 50% for bases with Q ≤ 20. The copyright of the Perl script belongs to Metware (http://www.metware.cn/). Additionally, Q20, Q30, and GC content were calculated. Ribosomal reads were removed using Bowtie [74]. Then, cleaned readswere mapped to the flowering Chinese cabbage genome (http://39.100.233.196:82/download_genome/Brassica_Genome_data/Brara_Chiifu_V3.0/)
Reviewer Suggestion; For raw reads, please provide the bioproject number and SRR numbers for the reads. The SUB number is irrelevant.
Author Response; thanks for your kind suggestion and i am sorry for my mistake. Here we have upload our RNA seq. data to NCBI under accession no. SRR, as given bellow and highlighted in original MS;
Accession numbers
The transcriptome data were submitted into NCBI’s SRA (sequence Read Archive) database using accession no. SRR28051825, SRR28051824, SRR28051823, SRR28051822, SRR28051821, SRR28051820, SRR28051819, SRR28051818, SRR28051817, SRR28051816, SRR28051815, SRR28051814, SRR28051813, SRR28051812, SRR28051811, SRR28051810, SRR28051809, SRR28051808, SRR28051807, SRR28051806, SRR28051805, SRR28051804, SRR28051803, SRR28051802, SRR28051801, SRR28051800, SRR28051799.

Round 4
Reviewer 2 Report
Comments and Suggestions for Authors
Thank you for providing all informations that I've asked for.